# Clinical Outcome and 8-Year Follow-Up of Alveolar Bone Tissue Engineering for Severely Atrophic Alveolar Bone Using Autologous Bone Marrow Stromal Cells with Platelet-Rich Plasma and β-Tricalcium Phosphate Granules

**DOI:** 10.3390/jcm10225231

**Published:** 2021-11-10

**Authors:** Izumi Asahina, Hideaki Kagami, Hideki Agata, Masaki J. Honda, Yoshinori Sumita, Minoru Inoue, Tokiko Nagamura-Inoue, Arinobu Tojo

**Affiliations:** 1Division of Stem Cell Engineering, The Institute of Medical Science, The University of Tokyo, Tokyo 108-8639, Japan; hideki.agata@gmail.com (H.A.); honda-m@dpc.agu.ac.jp (M.J.H.); y-sumita@nagasaki-u.ac.jp (Y.S.); 2Department of Regenerative Oral Surgery, Unit of Translational Medicine, Nagasaki University Graduate School of Biomedical Sciences, Nagasaki 852-8523, Japan; 3Tissue Engineering Research Group, Division of Molecular Therapy, The Advanced Clinical Research Center, The Institute of Medical Science, The University of Tokyo, Tokyo 108-8639, Japan; inouedental20190501@gmail.com (M.I.); aritojo@g.ecc.u-tokyo.ac.jp (A.T.); 4Department of Oral and Maxillofacial Surgery, Matsumoto Dental University, Shiojiri 399-0781, Japan; 5Department of Oral Anatomy, Aichi-Gakuin University School of Dentistry, Nagoya 464-0821, Japan; 6Department of Cell Processing and Transfusion, The Institute of Medical Science, The University of Tokyo, Tokyo 108-8639, Japan; tokikoni@ims.u-tokyo.ac.jp

**Keywords:** bone regeneration, tissue engineering, mesenchymal stromal cells, bone marrow stromal cells, mesenchymal stem cells, clinical study, alveolar bone regeneration

## Abstract

Background: Although bone tissue engineering for dentistry has been studied for many years, the clinical outcome for severe cases has not been established. Furthermore, there are limited numbers of studies that include long-term follow-up. In this study, the safety and efficacy of bone tissue engineering for patients with a severely atrophic alveolar bone were examined using autogenous bone marrow stromal cells (BMSCs), and the long-term stability was also evaluated. Methods: BMSCs from iliac bone marrow aspirate were cultured and expanded. Then, induced osteogenic cells were transplanted with autogenous platelet-rich plasma (PRP) and β-tricalcium phosphate granules (β-TCP) for maxillary sinus floor and alveolar ridge augmentation. Eight patients (two males and six females) with an average age of 54.2 years underwent cell transplantation. Safety was assessed by monitoring adverse events. Radiographic evaluation and bone biopsies were performed to evaluate the regenerated bone. Results: The major population of transplanted BMSCs belonged to the fraction of CD34^−^, CD45^dim^, and CD73^+^ cells, which was only 0.065% of the total bone marrow cells. Significant deviations were observed in cell growth and alkaline phosphatase activities among individuals. However, bone regeneration was observed in all patients and the average bone area in the biopsy samples was 41.9% 6 months following transplantation, although there were also significant deviations among each case. No adverse events related to the transplants were observed. In the regenerated bone, 27 out of 29 dental implants were integrated. Dental implants and regenerated bone were stable for an average follow-up period of 7 years and 10 months. Conclusions: Although individual variations were observed, the results showed that bone tissue engineering using BMSCs with PRP and β-TCP was feasible for patients with severe atrophic maxilla throughout a long-term follow-up period and was considered safe. However, further studies with a larger number of cases and controls to confirm the efficacy of BMSCs and the development of a protocol to establish a reproducible quality of stem cell-based graft material will be required.

## 1. Introduction

Although the number of patients who wish to have oral rehabilitation with dental implants has increased, many of them do not qualify because of insufficient alveolar bone volume. For these patients, an autologous bone graft is considered the gold standard and the procedure has been widely performed. Although autologous bone transplantation is the most predictable and reliable treatment to regenerate bone, it requires a donor site and may cause morbidity. Bone graft substitutes including allografts, xenografts, and artificial bone substitutes, such as hydroxyapatite or tricalcium phosphate, represent other options. These bone substitute materials alone can be applied to small or cystic bone defects which are surrounded by bony walls; however, they cannot repair severely atrophic alveolar bone since these materials have only osteoconductivity and not osteoinductivity [1]. Recently, to add osteoinductivity to bone substitutes, the use of various types of active biomolecules, including growth factors, peptides derived from extracellular matrix proteins, and small molecules that influence bone mass, have been evaluated in clinical trials [2]. Among them, recombinant human bone morphogenetic protein-2 (rhBMP-2) [3] and platelet-derived growth factor-BB (PDGF-BB) [4] have become commercially available and used in dentistry. Although biological reagents have great potential, there are still some hurdles, such as high cost, instability [5,6], and potential side effects, such as ectopic bone formation, osteolysis, and soft tissue swelling [7]. These novel methods are not a complete substitute for autologous bone grafts, since it is difficult to adopt an optimal dose and timing for growth factor application, and biological tissue regeneration is regulated by not one, but a variety of growth factors and bioactive molecules.

Accordingly, cell-based therapies and several clinical studies have been carried out using a variety of cell sources for bone regenerative therapy [2]. In terms of alveolar bone regeneration, bone marrow stromal cells (BMSCs) [8,9], periosteum derived cells (PDCs) [10,11], and adipose stem cells (ASCs) [12,13] have been used as stem/progenitor cells. ASCs are known to contain multilineage differentiation potential with a 10-fold higher colony forming unit-fibroblast than BMSCs [14]. However, liposuction for harvesting adipose tissue faces resistance in the dental field and a strong osteoblastic inducer is required to promote differentiation into osteoblastic cells. In contrast, the periosteum contains abundant osteoprogenitor cells in the inner cambium layer [15]. However, fibroblastic cells from outside the cambium layer tend to proliferate; thus, a method of selective proliferation of osteoprogenitor cells has not yet been established. Among them, BMSCs have been used in most studies and are considered a promising cell source for bone tissue engineering, because BMSCs are known to contain both pluripotent stem cells and osteoprogenitor cells. It is relatively easy to harvest tens of milliliters of bone marrow aspirate under local anesthesia and it is also easy to proliferate BMSCs on culture dishes [16,17].

To regenerate tissue for cell-based therapy, three essential components, the so-called tissue engineering triad, are required, including progenitor cells, stimulatory factors, and a cell scaffold [18]. In terms of stimulatory factors, the above-mentioned bioactive molecules, such as BMPs, PDGFs, and other factors, including growth and differentiation factor and parathyroid hormone, are utilized [2]. In addition, platelet-rich plasma (PRP) could be a candidate because PRP is known to accelerate bone regeneration and is easy to prepare [19]. In terms of the scaffold in bone tissue engineering, calcium phosphate materials, such as hydroxyapatite (HA) and β-tricalcium phosphate (β-TCP), are widely utilized because they are inorganic components of bone tissue. They also meet the requirements of the scaffold for constructing a 3D structure to shape the augmented bone and enable cells and stimulatory factors to attach. Next, we selected BMSCs as stem/progenitor cells, β-TCP for the scaffold, which is absorbable to the regenerated bone, and PRP as a stimulatory factor.

Although there are several clinical trials based on studies of bone tissue engineering using animal cells and models, information regarding the clinical usefulness of these strategies is limited; the standard method for alveolar bone regeneration has not been established [20,21], and there is no information regarding the long-term results of bone tissue engineering. Therefore, we assessed the feasibility, safety, and efficacy of alveolar bone tissue engineering using autologous BMSCs with PRP and β-TCP for severely atrophied alveolar bone. We also evaluated the results 8 years postoperatively. The accumulation of clinical data and identification of potential problems will lead to the development of a standard method for alveolar bone tissue engineering.

## 2. Materials and Methods

This study conformed to the tenets of the Declaration of Helsinki and the protocol was approved by the institutional review board of the Institute of Medical Science, The University of Tokyo (IMSUT) (clinical study, No. 16-22; long-term follow-up study, No. 25-21). All subjects provided written informed consent. This study was registered in the UMIN Clinical Trials Registry (UMIN000045309).

### 2.1. Inclusion Criteria

Patients who expected to have dental implant treatment.Patients who had more than two continuous tooth defects in which fixed prostheses were not applicable.Patients who showed a severely atrophic maxilla or mandible, which required bone transplantation.(1)The width of the alveolar bone ridge at the installation sites was less than 5 mm.(2)In the maxilla, the distance between the alveolar ridge and the sinus floors was less than 5 mm.(3)In the mandible, the distance between the ridge and mandibular canal was less than 5 mm.Good oral hygiene was maintained.Aged 20 years or older, but younger than 70 years.An understanding of the informed consent form and provided consent for the study.

### 2.2. Exclusion Criteria

Diabetes and/or autoimmune diseases.Hemorrhagic diathesis in which partial thromboplastin time (PT) was lower than 50% and activated partial thromboplastin time (APTT) was less than 23.5 or longer than 42.5 s.Uncontrollable infectious diseases.Osteoporosis.Liver dysfunction with aspartate aminotransferase (AST) values less than 10 or more than 40 IU/L, or with alanine aminotransferase (ALT) values less than 5 or more than 45 IU/L.Pregnant or possible pregnancy.Allergy to any medications used in this study and/or the presence of an allergy that requires continuous systemic medication.Other special conditions that the responsible physician considered inappropriate.

### 2.3. Number of Subjects and Duration of Study

Ten patients were enrolled in this study; it started with a limited number of participants as a phase I/II pilot study primarily to assess the feasibility and the follow-up period was two years after cell transplantation for an initial step.

### 2.4. Donor Screening

As a screening test for the subjects before enrollment in this study, serum was tested for HBs antigen (CLIA), HBs antibody (CLIA), HBc antigen (CLIA), HCV antibody (RIA solid phase), HIV antigen and HIV antibody (ELISAs), syphilis serology (RPR), syphilis serology (TPHA) and HTLV-1 antibody (CLIA). The subjects were negative for all tests. Biochemical markers, white blood cells, red blood cells, and platelet counts, hemoglobin, hematocrit, PT, APTT, fibrinogen, MCV, MCH, and MCHC were also checked.

### 2.5. Autologous Serum Preparation

Prior to bone marrow aspiration, 200–400 mL of peripheral blood was collected and an autologous serum was prepared. Peripheral blood was stored at 4 °C for 1 h and then centrifuged. Serum was transferred to fresh bags (approximately 50 g each) for aseptic cryopreservation (Nipro Corp., Osaka, Japan). Except for one case in which the serum was used immediately, the bags were stored in a freezer at −20 °C until use. Before use, the serum was thawed and added to culture medium containing antibiotics to a final serum concentration of 10%.

### 2.6. Harvest and Expansion of Bone Marrow Stromal Cells

Bone marrow was harvested from the iliac bone crest under local anesthesia. Marrow (10 mL) was aspirated into a syringe containing 500 U of heparin. Approximately 0.5 mL of bone marrow aspirate was reserved for flow cytometric analyses. The remainder of the aspirated bone marrow was diluted four-fold with α-MEM (Gibco BRL, Grand Island, NY, USA) containing 10% autoserum, 2.5 µg/mL of amphotericin B, and 50 µg/mL of gentamicin sulfate. The cells were plated in 150 cm^2^ flasks (Corning, New York, NY, USA). Non-adherent cells were discarded at the time of medium change. Adherent cells were cultured as BMSCs [22] and maintained in the same medium at 37 °C and a 5% CO_2_ atmosphere. The cells demonstrated a typical spindle shape, which was maintained throughout the cell culture period. When cells attained 80% confluency, they were subcultured into flasks and expanded to three or four T-150 flasks. For subculture, the cells were first washed with phosphate-buffered saline (PBS) then treated for 10 min at 37 °C with TrypLE^TM^ select (Invitrogen, Carlsbad, CA, USA), washed with PBS, resuspended in the same fresh culture media, and seeded into T-150 flasks. Cells at passage 2 were induced with osteogenic induction medium for 7 days. The osteogenic induction medium contained 10 nM of dexamethasone (Dex, Sigma-Aldrich, St. Louis, MO, USA), 100 μM of ascorbic acid (Wako Pure Chemical Industries, Ltd., Osaka, Japan) in α-MEM containing 10% autoserum, 2.5 µg/mL of amphotericin B, and 50 µg/mL of gentamicin sulfate. The medium was changed twice per week.

#### 2.6.1. Alkaline Phosphatase Activity Analysis

Osteogenic differentiation of the BMSCs was confirmed by alkaline phosphatase (ALP) activity as previously described [23]. Briefly, 1 × 10^6^ harvested cells were incubated with 400 μL of 20 mM HEPES (Dojindo, Kumamoto, Japan) buffer (pH 7.5) containing 1% Triton X-100 (Wako Pure Chemical Industries, Ltd., Osaka, Japan) to extract proteins. After extraction, the supernatants were incubated with BCA protein assay reagent (Pierce Thermo Scientific, Waltham, MA, USA) or p-nitrophenyl phosphate solution (Sigma-Aldrich, St. Louis, MO, USA) in tubes (30 min for the protein assay and 5 min for p-nitrophenyl phosphate, in a dark room at room temperature). The conversion of p-nitrophenyl phosphate into p-nitrophenol was terminated by adding 3N NaOH after 5 min. The absorbance (405 nm for p-nitrophenol analysis; 570 nm for protein analysis) was measured using a spectrophotometer (Bio-RAD Laboratories, Inc., Hercules, CA, USA). ALP-specific activity was expressed as μmol p-nitrophenol/min/μg protein.

#### 2.6.2. Safety Tests

Culture medium with 10% autologous serum (1 mL) underwent three safety checks. To test for contamination by bacteria and fungi, the medium was subjected to aerobic and anaerobic culture, which was performed every two weeks and prior to cell transplantation.

Before cell transplantation, the medium was also tested for mycoplasma contamination. DNA was extracted from the prepared medium using phenol: chloroform: isoamyl alcohol (PCI) (Sigma-Aldrich, St. Louis, MO, USA). Equal volumes of PCI were added to the medium (600 µL) and centrifuged at 15,000 rpm for 5 min at room temperature. The supernatant was mixed with ice-cold 100% isopropanol (Wako Pure Chemical Industries, Ltd., Osaka, Japan) (400 µL) and centrifuged at 15,000 rpm for 10 min. The pellet was then rinsed with ice-cold 70% ethanol (Wako Pure Chemical Industries, Ltd., Osaka, Japan) (400 µL) and centrifuged at 15,000 rpm for 5 min. The final pellet was dried for 3 min and then dissolved in distilled water (20 µL). A two-step PCR reaction was performed [24], which contained 1 μL of each primer (10 pmol/μL), 5 μL of 10x reaction buffer, and 10 nmol of each deoxynucleotide, template, and water to a volume of 49 μL. The primers for the first-step PCR were MCGpF11 (5′-ACACCATGGGAG(C/T)TGGTAAT-3′) and R23-JR (5′-CTCCTAGTGCCAAG(C/G)CAT(C/T)C-3′). The second-step PCR was also performed in a 50 µL volume with internal primers R16-2 (5′-CTG(C/G)FF(A/C)TGGATCACCTCCT-3′) and MCGpR21 (5′-GCATCCACCA(A/T)A(A/T)AC(C/T)CTT-3′). The mixture contained 1 μL of the first PCR product. Thirty-five cycles were run using the following conditions: denaturation at 94 °C for 30 sec, annealing at 55 °C for 2 min, and polymerization at 72 °C for 2 min. Amplified DNA was separated on a 1.5% agarose gel and soaked in TAE buffer containing 0.1 µg/mL of ethidium bromide. The products were analyzed using a ChemiDoc XRS gel documentation system (Bio-RAD Laboratories, Inc., Hercules, CA, USA).

Five microliters of culture medium were diluted 100-fold in normal saline solution and tested by the Endosafe PTS system (Charles River Laboratories, Inc., Wilmington, MA, USA) for endotoxins before cell transplantation. Samples containing greater than 1.0 EU/mL were considered positive.

#### 2.6.3. Flow Cytometry

Flow cytometric analysis was performed with a FACS Aria flow cytometer (Becton Dickinson, San Jose, CA, USA). The following antibodies were used: phycoerythrin-conjugated (PE-) and allophycocyanin-conjugated (APC-) antibodies against CD73 and CD34, respectively (BD Pharmingen, Franklin Lakes, NJ, USA), and a biotinylated antibody against CD45 (Invitrogen, Carlsbad, CA, USA). The biotinylated antibody was detected with streptavidin Pacific Blue (Molecular Probes–Invitrogen, Carlsbad, CA, USA). Propidium iodide (Dojindo, Kumamoto, Japan) was used to detect dead cells. As a negative control, each color-conjugated mouse-IgG1k (BD Pharmingen, Franklin Lakes, NJ, USA) was utilized. Data analysis was performed using FlowJo software (TreeStar, Inc., San Carlos, CA, USA).

### 2.7. Preparation of Transplants

On the day of transplantation, 60 mL of peripheral blood was collected to generate autologous PRP. PRP was prepared using a kit (Smart Prep, Harvest Technologies Corp., Plymouth, MA, USA) according to the manufacturer’s protocol. To harvest cells, cells were detached with TrypLE^TM^ Select (Invitrogen, Carlsbad, CA, USA), counted with a hemocytometer, and 1 × 10^6^ harvested cells were used for the ALP assay. Another 1 × 10^6^ cells were frozen for future analyses. The remainder of the cells were transferred to the operation room on ice and mixed with autologous PRP. The PRP/cell mixture was then combined with autologous thrombin made with the same kit and 10% CaCl_2_ to produce a gel. The gel was mixed with a 40% volume of β-TCP granules (Olympus Terumo Biomaterials Corp., Tokyo, Japan) and transplanted.

### 2.8. Surgical Procedure

The transplantation was performed under local anesthesia and intravenous sedation with propofol. The lateral sinus wall was opened and the sinus floor mucous membrane was carefully separated and elevated inward with the bone fragment. The transplant was filled into the space between the sinus floor and the elevated membrane. The mucoperiosteal flap was repositioned and sutured. In the case of alveolar ridge augmentation, the mucoperiosteal flap was elevated and the transplant was placed on the atrophic alveolar ridge where the dental implant installation was planned and covered with a Goretex membrane (Goretex^®^ TR Membrane, Japan Goretex Co. Ltd., Tokyo, Japan). The membrane was fixed with screws. After extension of the flap, the incision was sutured.

In both the sinus floor elevation and alveolar ridge augmentation cases, dental implants were installed six months after transplantation. In the cases of alveolar ridge augmentation, the membrane was removed prior to implant installation.

### 2.9. Evaluation

Panoramic X-rays were evaluated before the operation and at 6, 12, and 24 weeks, and 1 and 2 years after the operation. Computed tomography (CT) was performed before the operation, at 6 months, and 1 and 2 years after the operation. The amount of regenerated bone was calculated using SimPlant software (Materialize, Leven, Belgium). A bone biopsy was performed at 24 weeks after cell transplantation at the time of dental implant installation using a trephine bur (2 mm inner diameter and 3 mm outer diameter; Stoma am Mark GmbH, Emmingen-Liptingen, Germany). After embedding in resin, non-decalcified ground sections were prepared and the sections were evaluated with Villanueva bone staining, Villanueva–Goldner staining, or hematoxylin and eosin staining (H&E).

#### Histomorphometric Analysis

Light microscopy images were captured with a digital camera (Carl Zeiss AG, Oberkochen, Germany) and transferred to a computer. The extent of new bone area, the area of remaining scaffold, the area of fibrous tissue, and the area of bone marrow-like tissue were manually assessed using ImageJ software (Scion Corporation, Frederick, MD, USA) by an examiner. The size of these specific areas was expressed as the percentage of the total area of the section.

### 2.10. Long-Term Follow-Up

We conducted long-term follow-up observation for 5 out of 8 subjects, who could be contacted and provided the additional written informed consent 7 years after transplantation. The survival of the dental implants installed into the regenerated alveolar bone was evaluated and imaging analysis including panoramic X-ray and CT was performed.

## 3. Results

### 3.1. Cases

Information for the study subjects is summarized in Table 1. Ten patients (two males and six females) were enrolled and received bone marrow aspiration. In one case, the total cell number did not reach the required level for the protocol. In another case, bacterial contamination of the serum was suspected, so the cells were discarded. Since those patients refused a second aspiration, they were dropped from the study. Cell transplantation was performed in eight cases (two males and six females) with an average age of 54.2 years (Table 1). Sinus floor elevation was performed at nine sites in seven patients, and alveolar ridge augmentation was performed at four sites in three cases. Both procedures were performed in two cases.

### 3.2. Cell Fraction Analyses of Bone Marrow Aspirate

To investigate the fraction of BMSCs from the whole bone marrow aspirate, flow cytometry was performed with anti-CD34 (Figure 1a), CD45 (Figure 1a,b), and CD73 (Figure 1b) antibodies. The major population of BMSCs belonged to the fraction of CD34^−^, CD45^dim^, and CD73^+^ cells, which accounted for only 0.065% of the total bone marrow cells. This was much lower than the putative fraction of hematopoietic stem cells (CD34^-^ and CD45^-^), which was 0.89% (Figure 1a,b). Although the characteristics of the cultured BMSCs were not defined with the samples used in this study, the results from our previous study using the same protocol indicated that the dominant cell type was positive for CD10, CD29, CD73, CD90, CD146, and HLA-ABC, and negative for CD3, CD14, CD19, CD34, and CD45 [18].

### 3.3. Cell Growth and Diferentiation

The average cell number at the time of transplantation was 1.6 × 10^7^ cells; however, the cell proliferating capability varied among individuals. Similar findings were observed in terms of cell differentiation. Although the exact same induction protocol was used, the final harvested cell number and the levels of ALP activity varied among individuals (Figure 2a,b).

### 3.4. Clinical Findings

In the case of sinus floor augmentation (SFA), the created space was filled with the transplant consisting of PRP gel with BMSCs and β-TCP granules as described above (Figure 3a). Six months after cell transplantation, the mucoperiosteal flap was re-elevated (Figure 3b). At this time, regenerated bone was observed. Drilling was performed to the maxilla and the fixtures were installed (Figure 3c). Although dental implants were installed in all cases, the texture and hardness of the regenerated bone varied. For ARA, subjects with severely atrophic alveolar ridges were recruited (Figure 4a) and guided bone regeneration was performed (Figure 4b). Six months after cell transplantation, the mucoperiosteal flap was re-elevated (Figure 4c). The surface was covered with relatively hard bone. Drilling was performed and dental implants were installed (Figure 4d). The hardness of the regenerated bone also varied.

### 3.5. CT of Alveolar Bone

CT images from representative cases are shown in Figure 5. CT was performed before the operation (Figure 5a,d) and at 6 months (Figure 5b,e) and 12 months (Figure 5c,f) after cell transplantation. In the cases of SFA (Figure 5a–c) and ARA (Figure 5d–f), the borderline between the transplants and original bone was still visible at 6 months (Figure 5b,e). However, it became almost continuous after 12 months (Figure 5c,f).

### 3.6. Time Course of the Volume of Regenerated Bone

The volume of regenerated bone was analyzed using CT data and software at 6, 12, and 24 months after cell transplantation (Figure 6a). Generally, the volume decreased over time, although the time course varied among individuals. The average volume of regenerated bone was 75.2% at 1 year and 62% at 2 years after transplantation compared with that at 6 months (Figure 6b).

### 3.7. Histology of Regenerating Bone

Bone biopsy was performed at 6 months at the site of the implant installation using a trephine bur. The histology from representative cases is shown in Figure 7. Non-decalcified sections were stained with H&E (Figure 7a), Villanueva bone staining (Figure 7b), and Villanueva–Goldner staining (Figure 7c). Although bone formation was observed in all cases, the area of new bone formation and remaining scaffold varied among patients. Bone formation was mostly observed adjacent to the scaffold (Figure 7a,b). Bone marrow-like tissue was not observed except in one case. In the case of ARA, relatively mature bone was usually observed on the surface (Figure 7c).

### 3.8. Histomorphometric Analyses

The average of new bone area was 41.9 ± 27.8% at 6 months after cell transplantation, though there was variation among individuals (Figure 8a). The area of remaining scaffold was less than 40% and was almost invisible in two out of eight subjects (Figure 8b). The area of fibrous connective tissue-like structure was also determined. This was a major component of the regenerated bone in some of the samples and generally inversely related to the amount of new bone formation (Figure 8c). Bone marrow-like tissue was not observed, except in one patient (Figure 8d).

### 3.9. Dental Implants

In total, 29 dental implants were installed to the regenerated bone and 27 were integrated (93%), whereas another two implants were removed at the time of abutment connection. There was no infection or other pathological changes in cases of unintegrated implants. During the follow-up period, no trouble was noted for the integrated implants (~29 months).

### 3.10. Safety Aspect of the Therapy

During the treatment and follow-up period (45–66 months, average of 56 months after cell transplantation), no side effects or health concerns were noted.

### 3.11. Long-Term Follow-Up

All 20 dental implants installed into the regenerated alveolar bone of five subjects survived an average of 94 months (86–107 months). CT images showed that the regenerated bone was indistinguishable from the host bone (Figure 9a). The average bone volume at 94 months after cell transplantation, calculated from CT images, was decreased to 40% and 59% compared with that at 6 and 24 months, respectively (Figure 9b). There was variation among individuals, but the amount of decreased volume was smaller in the ARA cases than in the maxillary SFA cases.

## 4. Discussion

In this study, bone regeneration was achieved and dental implant installation was feasible in all subjects who suffered from severely atrophic maxilla and required bone transplantation. In particular, bone marrow aspiration could be performed under local anesthesia within 30 min and no adverse events were observed. Since harvesting autologous bone is inevitably associated with complications [25], bone tissue engineering with BMSCs is advantageous as a less invasive treatment. The safety and clinical efficacy of alveolar bone tissue engineering were examined in the present study. During the treatment and follow-up period, no complications related to cell transplantation were observed. This confirmed previous reports on bone tissue engineering using BMSCs in which no complications were reported [8,9,21,26,27]. In the present study, five of eight patients were followed up with for over 8 years after transplantation and showed stable regenerated alveolar bone and dental implants. Although our findings were from a limited number of cases, long-term stability of the regenerated bone has been presented.

In terms of efficacy, we observed bone regeneration in eight out of eight cases. The results from the histomorphometric analyses showed the average bone area was 41.9% at 6 months after transplantation. Although comparison with other studies may not be possible, the reported new bone areas after an autologous bone graft for sinus floor elevation were almost identical to that in this study. For example, when autologous bone was transplanted to the alveolar ridge or sinus floor, the average new bone area was between 31.2%–37.7% [28,29,30,31]. When autologous bone was transplanted with calcium phosphate, the new bone area was 44.24 +/− 13.79% [32]. In the cases of autologous bone transplantation, absorption is a serious problem. However, only limited data are available regarding the stability of tissue-engineered bone. A comparable study using an autologous bone graft to the sinus revealed that the graft height was reduced to 67.8–80.7% after 1 year and 55.8%–72.2% after 5 years [33]. Although direct comparison is not possible, absorption does occur to the present tissue-engineered bone similar to that of autologous bone grafts. This absorption rate may be dependent on the property of the scaffold material. It is reported that 90% of the engineered bone from human cells seeded into the resorbable, polyglycolic-polylactic acid scaffold was resorbed [34]. Our results show that the decreasing rate of the augmented bone in the first two years was larger than that the next 6 years (Figure 9). It seems that the β-TCP used in this study was replaced with host bone within two years, after which the regenerated bone may remodel with natural bone.

One interesting finding of this study was the significant differences observed by CT images at 6 months and 1 year after cell transplantation. At six months, the borderline between transplants and the surrounding bone was still clear in most cases. However, the borderline became almost invisible at 12 months. This may reflect the amount of remaining β-TCP granules at this stage. Histological analyses showed that bone regeneration occurred 6 months after cell transplantation. The degradation process of β-TCP continued until 12 months and the maturation of regenerated bone gradually occurred during this period. It is of interest to know whether the bone formed after 12 months was close to normal bone tissue. CT analyses suggest that it is similar to normal bone tissue. No fundamental differences were observed in the bone regeneration process between the sinus floor elevation and ARA.

The percentages of new bone area appeared to correlate with the physical strength of the regenerated bone, thus affecting the initial stability of the dental implant. However, even patients whose regenerated bone showed less new bone area and minimum implant stability at 6 months had successful dental implant integration at the time of abutment connection (6 months after implant installation). In the early bone regeneration cases, the bone texture after 6 months was close to that of native bone. In the cases with delayed bone regeneration, it is reasonable to speculate that osteoconduction, rather than osteogenesis or osteoinduction, occurred. It is of interest to know the difference of origin of the osteoblasts in these cases. In the present study, two implants were not integrated. Initial stability of the dental implants was sufficient, and infection was not evident, but the implant failure may have resulted from differences in bone quality or the implant placement procedure.

As shown in this study, the fraction of BMSCs in whole bone marrow was relatively small (>0.065%), and in vitro expansion was necessary to obtain a sufficient number of cells for transplantation. Based on the expression of cell surface markers, BMSCs expanded with our protocol were identical to so-called mesenchymal stem cells or multipotent mesenchymal stromal cells (MSCs) as reported previously [35,36]. However, there were individual variations in in vitro cell character and in vivo bone quality among patients, which may affect the effectiveness of this treatment. ALP activity and proliferation of the cells were not related to in vivo bone forming ability and these variations were not related to patient gender, age, or the site of implantation. This is in accordance with a previous study by Meijer et al., in which they showed that ALP-stained cell constructs did not always induce bone formation in patients and stated the importance of a sufficient vascular supply [9]. Clinical efficacy could be affected by various factors, such as anatomical environment, surgical procedure, and transplant volume. Furthermore, the characteristics of the obtained cells differed among individuals, which may also affect efficacy. The osteogenic ability of BMSCs is easily lost after passage and this may also affect individual variation [22,35]. For successful treatment, it is essential to understand the reason for this individual variation and to establish parameters for consistent clinical efficacy.

Another limitation of the present study is the lack of a control group. It is possible that only the scaffold biomaterial, β-TCP, implantation could regenerate the atrophied alveolar bone by osteoconduction. It is also possible that PRP, but not the transplanted cells, accelerated the bone regeneration. We cannot rule out the possibility that the viability of the transplanted cells in vivo may cause variations in bone formation and the transplants without live cells in some cases. However, Kaigler et al. reported that both β-TCP with or without CD90+ stem cells prepared from bone marrow yielded sufficient bone height and volume in sinus floor elevation, but the regenerated bone in the stem cell group exhibited higher bone quality [37]. Thus, it may be worthwhile to use stem cells to achieve the high-quality bone regeneration as an alternative to autologous bone grafts. We used culture dish-adhered cells from bone marrow cells as BMSCs, whereas Kaigler et al. used a CD90+ stem and CD14+ monocyte-enriched cell population prepared by defined cell processing. This cell homogeneity may provide consistent results for bone regeneration compared with the present study. In contrast, Rickert et al. showed that bone marrow aspirates’ concentrated mononuclear cells without cell culture (which may provide a more heterogenous cell population) seeded on bovine bone mineral (BioOss^®^) induce a sufficient bone volume of new bone formation compared with that of autologous bone in SFA [38]. Furthermore, a recent clinical study using BMSCs with biphasic calcium phosphate as a scaffold showed successful results in the bone regeneration of severely atrophied mandible alveolar bone [39]. They prepared BMSCs with a similar method to ours, but supplemented with platelet lysate, whereas we added autologous serum to the culture medium. The variation in supplemental serum performance may cause the individual deviation of BMSC features in the present study. On the other hand, another clinical study used BMSCs, which are CD90+ enriched stem cells, successfully for sinus floor elevation and alveolar ridge preservation [40], which resulted in limited bone regeneration of large alveolar bone defects [41]. Taken together, the safety of stem cell therapy is confirmed; however, further investigation including cell preparation methods and choice of scaffold material is necessary to establish this technique as a standard therapy for large alveolar bone regeneration.

## 5. Conclusions

The results from this clinical study showed the feasibility of alveolar bone tissue engineering using autologous BMSCs. We did not observe any complications related to the transplanted cell constructs, which reflects the relatively safe nature of this treatment. However, the reason for individual variations was not determined. We could not identify the role of BMSCs in bone regeneration since there were large variations among individuals in both in vitro cell proliferation/differentiation and in vivo bone formation. Studies involving a larger number of cases with a control will further prove the safety and efficacy. A novel protocol, which enables more stable bone regeneration, should be considered in future clinical studies.

## Figures and Tables

**Figure 1 jcm-10-05231-f001:**
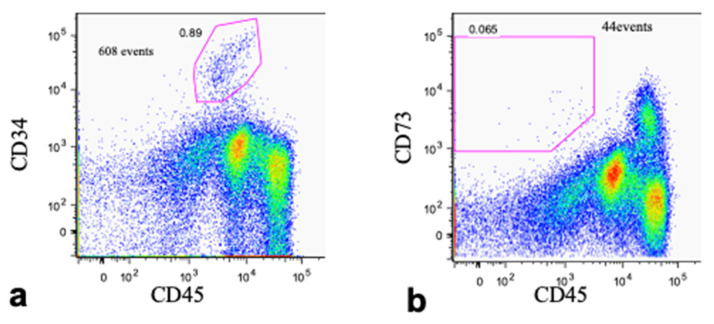
Flow cytometry of harvested whole bone marrow cells. The cells were sorted (**a**) with anti-CD34 and CD45 and (**b**) with anti-CD73 and CD45.

**Figure 2 jcm-10-05231-f002:**
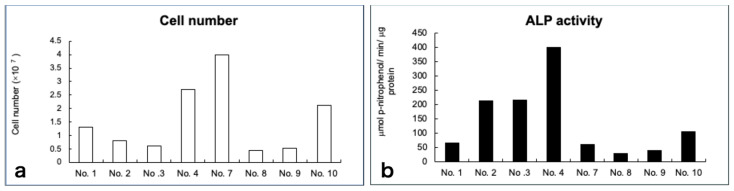
Total cell number (**a**) and alkaline phosphatase (ALP) activity (**b**) of the cultured cells at harvest before transplantation. ALP activity was expressed as μmol p-nitrophenol/min/μg protein.

**Figure 3 jcm-10-05231-f003:**
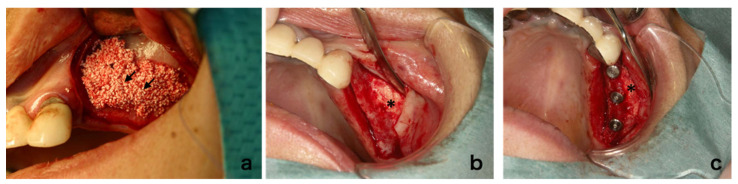
Maxillary sinus floor augmentation: (**a**) operative procedure (arrows indicate the transplanted material), (**b**) regenerated alveolar bone 6 months after surgery, (**c**) dental implant insertion (asterisk indicates regenerated bone).

**Figure 4 jcm-10-05231-f004:**
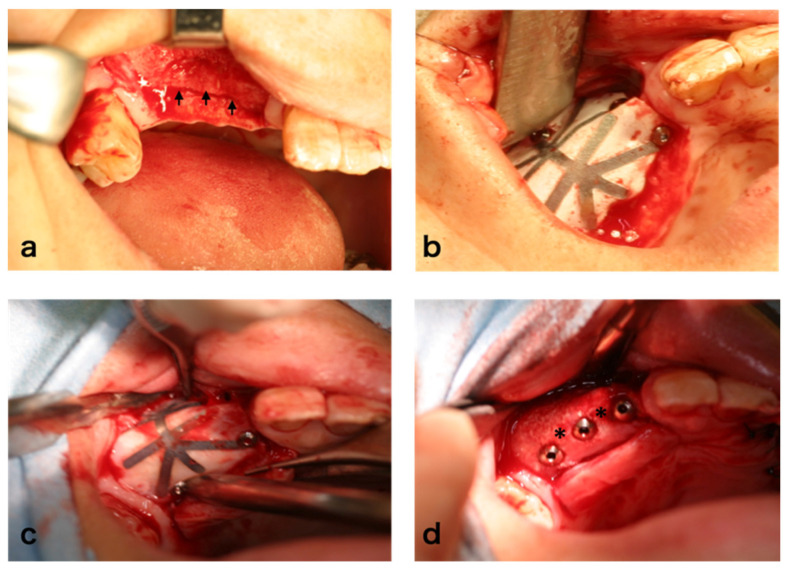
Alveolar ridge augmentation: (**a**) arrows show atrophied alveolar bone, (**b**) guided bone regeneration with ePTFE membrane, (**c**) 6 months after surgery, and (**d**) dental implant insertion (asterisks indicate regenerated bone).

**Figure 5 jcm-10-05231-f005:**
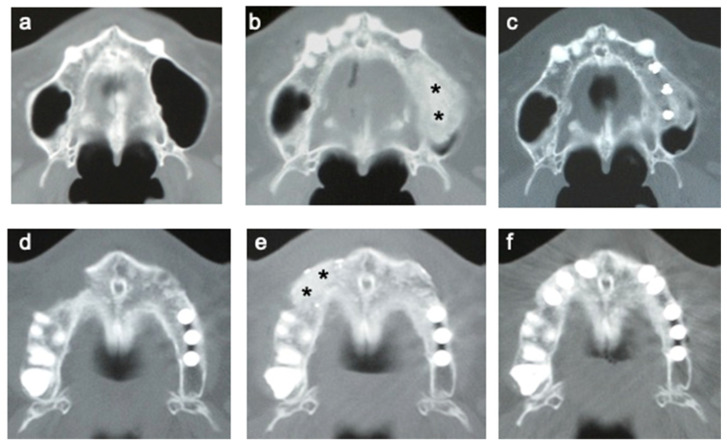
CT images of regenerated alveolar bone: (**a**–**c**) maxillary sinus floor augmentation and (**d**–**f**) alveolar ridge augmentation. (**a**,**d**) Before surgery, (**b**,**e**) 6 months after transplantation (asterisks indicate regenerated bone), and (**c**,**f**) 12 months after transplantation.

**Figure 6 jcm-10-05231-f006:**
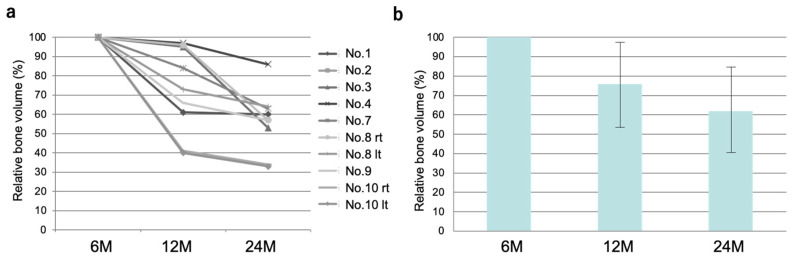
Time course of bone volume: (**a**) individual changes and (**b**) average bone volume.

**Figure 7 jcm-10-05231-f007:**
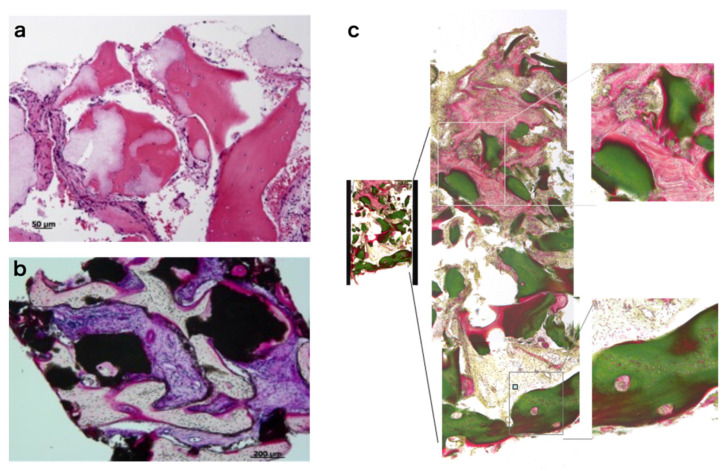
Histology of regenerated bone: (**a**) H&E staining, (**b**) Villanueva bone staining, and (**c**) Villanueva–Goldner staining.

**Figure 8 jcm-10-05231-f008:**
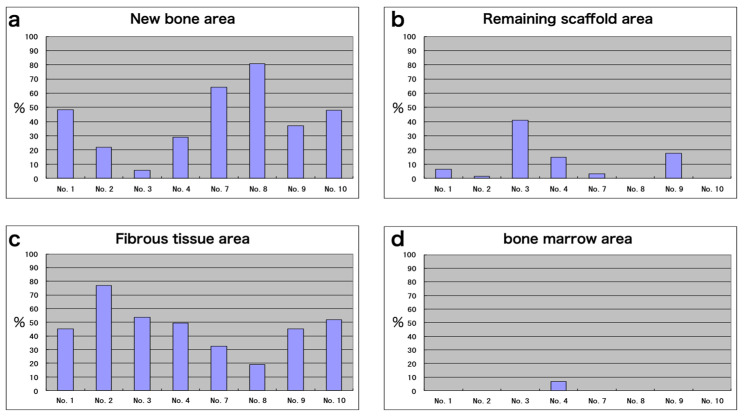
Histomorphometric analysis of bone biopsy samples at 6 months after transplantation: (**a**) new bone area, (**b**) remaining scaffold area, (**c**) fibrous tissue area, and (**d**) bone marrow area, expressed as a percentage of the total area.

**Figure 9 jcm-10-05231-f009:**
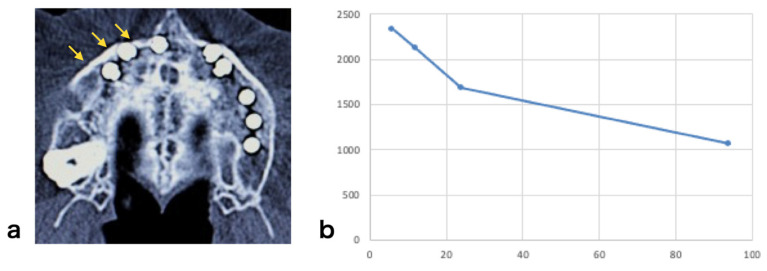
CT image and regenerated bone volume: (**a**) representative CT image of the regenerated alveolar bone with alveolar ridge augmentation and (**b**) time course of the regenerated bone volume.

**Table 1 jcm-10-05231-t001:** Summary of patient information.

No.	Age	Sex	Target Region and Procedure	Cell No. (×10^6^)	PRP (mL)	Plt. Count (10^4^/uL)	β-TCP (g)	No. of Implants
1	40	F	rt. SFA	13.3	2.8	ND	2.0	2
2	52	M	rt. SFA	8.2	3.0	ND	1.5	2
3	53	F	lt. SFA	6.0	3.5	73	2.0	3
4	52	F	lt. SFA	26.8	2.7	49.3	1.4	2
5	51	F	-	-	-	-	-	-
6	38	F	-	-	-	-	-	-
7	63	M	rt. SFA	39.6	4.0	57.3	1.0	4
8	57	F	bil. SFA 22~23 ARA	4.5	8.0	ND	3.0	6
9	60	F	15~12, 23~24 ARA.	5.2	4.0	ND	1.0	5
10	57	F	bil. SFA 24~25 ARA.	21.1	10.0	80.2	3.0	5

Ten patients were entered; however, patients no. 5 and 6 were dropped because of the risk of contamination in the serum and insufficient cell numbers, respectively. Sinus floor augmentation (SFA) and/or alveolar ridge augmentation (ARA) were performed in 7 and 3 cases, respectively. rt: Right side, lt: left side, bil: bilateral side, PRP: platelet-rich plasma, Plt.: platelet, β-TCP: tricalcium phosphate, ND: not detected.

## Data Availability

The data presented in this study are openly available in UMIN-CTR, ID: UMIN000045309.

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
