# Peer review of "Clinical Outcome and 8-Year Follow-Up of Alveolar Bone Tissue Engineering for Severely Atrophic Alveolar Bone Using Autologous Bone Marrow Stromal Cells with Platelet-Rich Plasma and β-Tricalcium Phosphate Granules"

_jcm, 2021, doi:10.3390/jcm10225231_

Round 1

Reviewer 1 Report

Title: write how long was the follow up.

abstract: it is written in a general way. Be more specific.

Introduction: Lots of flaws into your introduction. I can see that you did not respect the guidelines of writing a scientific paper.

Line 37-47: You don't have citations. 

Line 59: re-write your aim. 

M&M: I don't think you inclusion and exclusion criteria are alright. You are not specific. 

Line 89: 10 patients are insufficient to have a scientifically validity.

Figure 5: CBCT images are not clear. Insert more accurate.

Discussion: Needs to be re-written. There are still lots of english mistakes and you did not respect the scientific guidelines. Make comparisons with similar or different techniques.

Author Response

Thank you for taking time to review our manuscript. We appreciate your thoughtful and helpful comments and suggestions. Following those suggestions, we revised our manuscript with best of our efforts. Your comments were highly insightful and enabled us to improve the quality of our manuscript. Our point-by-point responses to each of your comments is the followings. Revised parts of the manuscript are high lightened with yellow maker.

Title: write how long was the follow up.

              The average follow up period was 7.8 years. Then, we add “8 years follow up” instead of “long term” in the title. (L2)

abstract: it is written in a general way. Be more specific.

              Thank you for the suggestion. We have intensively rewritten the abstract to be more specific. The addition of the sentences is high lighten with yellow maker. (L24-42)

Introduction: Lots of flaws into your introduction. I can see that you did not respect the guidelines of writing a scientific paper.

              We have almost completely revised the introduction following your comment. (L31-103)

Line 37-47: You don't have citations.

              We have added the citation of “Yamada, M., & Egusa, H. (2018). Current bone substitutes for implant dentistry. Journal of prosthodontic research62(2), 152–161.” (L55)

M&M: I don't think you inclusion and exclusion criteria are alright. You are not specific. 

              We are sorry that we could not well understand what’s point you indicated “we are not specific”. We cannot change the inclusion and exclusion criteria in this paper because the Internal Review Board had approved this study with those criteria. We itemize the criteria to make it more comprehensive. (L111-137)

Line 89: 10 patients are insufficient to have a scientifically validity.

              We agree with your comment, but this clinical study was a first-in-human study with this protocol and the primary endpoint was the feasibility as a phase I/II study. Then, we decided to start this clinical study with the limited number of participants, 10 patients. We revised as “Ten patients were enrolled into this study started with a limited number of participants as a phase I/II pilot study to assess the feasibility primarily and the follow-up period was two years after cell transplantation for an initial step.” (L139-141)

Figure 5: CBCT images are not clear. Insert more accurate.

              Thank you for your indication. We have changed the images of Figure 5. (L341)

Discussion: Needs to be re-written. There are still lots of english mistakes and you did not respect the scientific guidelines. Make comparisons with similar or different techniques.

              Thank you for your suggestion. We have intensively rewritten the discussion, especially paragraph 6 to make comparison with the other studies. We asked a professional English editing service to edit the manuscript. (L484-500)

Reviewer 2 Report

This is one of the rarest clinical studies with good editing and comprehensive analyses. However the following should be considered as the authors selected certain cases with atrophic ridges and used a combination therapy of autologous BMSCs, autologous PRP, and β-TCP granules and thus we can't' consider the clinical outcome relying only on BMSCs. The title and conclusion should be modified to be consistent with the transplant used. In addition, there are individual variations including gender, age, site of implantation and these should be considered when doing the statistical analysis.

Author Response

 Thank you for taking time to review our manuscript. We appreciate your thoughtful and helpful comments and suggestions. Following those suggestions, we revised our manuscript with best of our efforts. Your comments were highly insightful and enabled us to improve the quality of our manuscript. Our point-by-point responses to each of your comments is the followings. Revised parts of the manuscript are high lightened with yellow maker.

This is one of the rarest clinical studies with good editing and comprehensive analyses.

              Thank you for understanding the difficulty in carrying out this kind of clinical study.

However the following should be considered as the authors selected certain cases with atrophic ridges and used a combination therapy of autologous BMSCs, autologous PRP, and β-TCP granules and thus we can't' consider the clinical outcome relying only on BMSCs.

              Thank you for indicating a very critical point of this study. We agree with your comment. Then, we clearly stated the possibility that the outcome of this study could come from without BMSCs in the discussion. (L477-480)

The title and conclusion should be modified to be consistent with the transplant used.

              The title has been revised as “Clinical outcome and 8 years follow up of alveolar bone tissue engineering for severely atrophic alveolar bone using autologous bone marrow stromal cells with platelet rich plasma and b-tricalcium phosphategranules”. (L4-5)

In addition, there are individual variations including gender, age, site of implantation and these should be considered when doing the statistical analysis.

              According to your suggestion, we have re-evaluated the results, but we could not find any tendency related to gender, age, site of implantation mostly because of the limited number of cases. Then, we added the sentence of “and these variations were not related to the patients’ gender, age, and the site of implantation.” In the discussion. (L465-466)

Round 2

Reviewer 1 Report

All good. Thank you for your answers